# Facile Synthesis of Diatomite/β-Cyclodextrin Composite and Application for the Adsorption of Diphenolic Acid from Wastewater

**DOI:** 10.3390/ma15134588

**Published:** 2022-06-29

**Authors:** Min Hou, Zhiyi Wang, Qian Yu, Xianming Kong, Miao Zhang

**Affiliations:** 1School of Petrochemical Engineering, Liaoning Petrochemical University, Fushun 113001, China; hm189316@sina.com (M.H.); ww2474677557@sina.com (Z.W.); qyu@lnpu.edu.cn (Q.Y.); 2Department of Materials and Environmental Chemistry, Stockholm University, 10691 Stockholm, Sweden

**Keywords:** *β*-cyclodextrin, diatomite, diphenolic acid, adsorption, wastewater

## Abstract

Diphenolic acid (DPA) is a kind of endocrine-disrupting compound, which brings serious health problems to humans and animals. An eco-friendly and cost-effective adsorbent was prepared through a simple method, in which the *β*-Cyclodextrin(*β*-CD) was crosslinked onto the surface of diatomite (DA), the as-prepared DA/*β*-CD composite showed higher adsorption efficiency for DPA than DA as the host–guest interaction between DPA and *β*-CD. DA is a kind of biosilica with a hierarchical pore structure that provides enough surface area for the DA/*β*-CD. The surface area and pore size of DA/*β*-CD were investigated by nitrogen adsorption and desorption. The DA/*β*-CD composite illustrated a good adsorption capability, and was used for removing DPA from wastewater. The adsorption ratio of DPA could achieve 38% with an adsorption amount of 9.6 mg g^−1^ under room temperature at pH = 6. The adsorption isotherm curves followed the Langmuir (R^2^ = 0.9867) and Freundlich (R^2^ = 0.9748) models. In addition, the regeneration rate of the DA/*β*-CD was nearly at 80.32% after three cycles of regeneration. These results indicated that the DA/*β*-CD has the potential for practical removal of the EDC contaminants from wastewater.

## 1. Introduction

Endocrine-disrupting compounds (EDCs) are a series of pollutants that could bring estrogenic or androgenic problems to the public health. EDCs are usually present in water environments, such as surface wastewater, ground wastewater, and landfill leachate [1,2,3]. Various chemical substances from nature or artificial products have shown estrogen-like responses, including drugs, pesticides, and chemical intermediates [4].

Diphenolic acid (DPA) has been used as raw material to produce functional polymers [5]. DPA is usually prepared through the condensation between phenol (Pl) and levulinic acid (LA). The LA is cheap as that produced from cellulose-rich biomass [6], and Pl could be obtained from glucose [7]. Therefore, DPA can be regarded as a plant-derived compound and play an important role in the polymer industry. Thus far, more and more DPA-type polymer compounds have been developed, such as polyesters [8,9], flame retardants [10,11] and functionalized polymers [12]. Although DPA is regarded as a plant-derived compound and widely used in the polymer industry, discharging wastewater with DPA containing would bring potential hazards to the ecological environment. It is necessary to remove DPA from the environment. Currently, the elimination of DPA from water system mainly depend on photocatalytic reaction and oxidation methods. Although the photocatalysis is an efficient method to remove the DPA from water, the complicated process and high energy light were indispensable during the reaction process [13,14,15], which limits the practical application in DPA elimination. The oxidation methods include ozone oxidation [16], ferrate oxidation [17], and Fenton and Fenton-like oxidation methods [18,19], but these methods are time consuming and may bring pollution.

The physical adsorption methods were widely used for removing pollutants from water due to the high efficiency, low energy consumption, simple process, and low cost [20,21,22,23,24,25]. Functional materials with porous structures have attracted many interests for adsorbing specific compounds, as they could provide a huge surface area and excellent adsorption capability [26,27]. Tripathi et al. fabricated carbon materials with a hierarchically ordered micro–mesoporous structure, that was successfully used for adsorbing bisphenol A from the environment [28]. Zhang and coworkers prepared ordered mesoporous silica (SBA-15) with molecularly imprinted (MI) function, the MI-SBA-15 provided instant, stable, and high efficiency in organic pollutants removing [29].

Diatomite (DA) is a kind of biogenic deposit that has received extensive attention due to several unique properties, such as its high porosity, hydrophilicity, chemical stability, and low price. There are plenty of hydroxyl groups on the surface of DA that are beneficial for surface functionalization. DA composite has been widely used in catalysis, analysis, and environment [30,31]. 

Beta-cyclodextrin (*β*-CD) is a macrocyclic oligosaccharide molecule that is composed of seven units of D-glucopyranose [32]. *β*-CD has shown several merits, such as the non-polar hydrophobic cavity, polar hydrophilic surface, low price, non-toxicity, and biodegradability. The *β*-CD composite has been widely used as an absorbent for removing organic pollutants from the water system [33]. However, the hydrophilic feature of *β*-CD enables high solubility, limiting its application in water purification. Thus, numerous studies have focused on extending the application of *β*-CD. Ye et al. prepared DA/*β*-CD composite through emulsion polymerization under mild-conditions, which were applied in adsorbing methylene blue from wastewater. Nevertheless, epichlorohydrin (ECH) was needed during the fabrication process of DA/*β*-CD. The leakage of ECH would bring harmful effects on the environment and public health.

In this study, a novel functional composite (DA/*β*-CD) was synthesized through a simple and easy way. The *β*-CD derivative (Ts-*β*-CD) was firstly synthesized by cross-linking p-toluene sulfonyl chloride with *β*-CD. After that, Ts-*β*-CD was grafted on the surface of aminated DA. The schematic diagram of the preparation of DA/*β*-CD was shown in Figure 1. The designed and prepared DA/*β*-CD composite as adsorbent performed with a good adsorption capability, and the surface area and pore size of DA/*β*-CD were investigated by nitrogen adsorption and desorption. The porous structure of the DA and macrocyclic structure of *β*-CD enables the DA/*β*-CD composite to have the capability to rapidly adsorb endocrine-disrupting compounds (diphenolic acid, DPA) from wastewater.

## 2. Experimental Sections

### 2.1. Chemicals and Reagents

Diatomite (Celite 209) was purchased from Sigma-Aldrich (St. Louis, MI, USA), *β*-Cyclodextrin (*β*-CD, 98%), diphenolic acid (DPA, 98%+), ethyl acetate (EA, AR), Pyridine (AR, ≥99%), P-toluene sulfonyl chloride (PTSC, AR, 99%), N, N-Dimethylformamide (DMF, 99.9%) and (3-Aminopropyl) trimethoxy silane (APTMS, 97%) were purchased from Innochem (Beijing, China), hydrochloric acid (HCl), Sodium hydroxide (NaOH, AR), acetone, and ethanol were obtained from Sinopharm Chemical Reagent Co., Ltd. (Shanghai, China). All reagents were used without further purification. All working solutions were prepared with deionized water. The abbreviation of chemicals were listed in Appendix A.

### 2.2. Synthesis of Ts-β-CD

*β*-CD (6 g) and pyridine (34 mL) were added to a round bottom flask. The mixed solution was kept stirring at 2–4 °C, and then PTSC (0.8 g) was added to the mixed system within half an hour. The reaction was continued in an ice-water bath for eight hours, and then stirring was continued at room temperature for two days. The crude product was separated by rotary evaporation and washed with acetone, purified, and then Ts-*β*-CD was dried at 60 °C for six hours.

### 2.3. Synthesis of DA/β-CD

The DA (10 g) was firstly soaked in 50 mL ethanol solution of APTMS (0.5%) for five hours, then the DA was isolated and washed thoroughly with ethanol and dried in oven at 70 °C. After that, the aminated DA (2.5 g) and Ts-*β*-CD (1.5 g) were added to DMF (10 mL) solution, and the reaction system was kept stirring at 60 °C for seven hours, and the Ts-*β*-CD was successfully cross-linked onto the surface of DA. The products (DA/*β*-CD) were isolated by suction filtration and dried at 60 °C.

### 2.4. Characterizations

Scanning electron microscopic (SEM) images of DA and DA/*β*-CD were measured on SU 8010 field emission scanning electron microscope (Hitachi, Tokyo, Japan). UV–vis absorption spectra were obtained on Agilent Cary 5000 UV-vis-NIR spectrophotometer (Agilent, Palo Alto, CA, USA). Fourier transform infrared (FT-IR) spectrum of the DA, *β*-CD, and DA/*β*-CD were measured on the Nicolet 6700 spectrometer (PerkinElmer, Waltham, MA, USA). The NMR spectra were collected from a Bruker 400 NMR instrument (Bruker, Billerica, MA, USA). BET surface areas and porosities of the DA and DA/*β*-CD were measured by nitrogen adsorption and desorption with an Autosorb-IQ2-MP analyzer (Quanta chrome, Raleigh, NC, USA). The X-ray diffraction (XRD) pattern of samples were collected from Smart Lab x-ray diffractometer (Rigaku, Tokyo, Japan).

### 2.5. Adsorption Experiments

An aqueous solution of DPA was used as a simulated target system. The adsorption of DPA were carried out in beakers at room temperature, and the pH of the target system was adjusted by adding HCl (0.1 M) or NaOH (0.1 M). The adsorption capacity (q_t_, mg g^−1^) and removal efficiency (R%) were calculated using Equations (1) and (2), respectively, as follows: (1)$$q_{t}=\frac{V\left(C_{0}-C_{t}\right)}{m}\;$$
(2)$$R\%=\frac{C_{0}-C_{t}}{C_{0}}\times 100$$
where C_0_ is the initial concentration of DPA in the target water system; C_t_ is the concentration of DPA in the target water system at different times, V is the volume of the target water system, and m is the mass of the DA/*β*-CD composite used in the adsorption process. 0.2 g DA and DA/*β*-CD composites were placed in simulated wastewater (50 mL) with different concentrations of DPA. After that, the upper suspension was filtered with a syringe at certain time intervals, and the filtrate were transferred into cuvette for UV-vis spectra measurement. The adsorption results were the average values for three experiments.

Langmuir and Freundlich isotherm equations were used to fit the experimental results. Among them, the equations of Langmuir [34] and Freundlich [35] models are as follows:(3)$$\frac{C_{e}}{q_{e}}=\frac{1}{q_{m}b}+\frac{1}{q_{m}}C_{e}\;\mathrm{Langmuir}\;$$
(4)$${\log\;q}_{e}={\log\;k}_{F}+\left(\frac{1}{n}\right){\log\;C}_{e}\;\mathrm{Freundlich}\;$$
where, q_e_ is the adsorption capacity at the equilibrium of adsorption (mg g^−1^); b represents Lamgmuir constant in (L mg^−1^); q_m_ is the saturation adsorption capacity; k_F_ is the Freundlich constant in (L mg^−1^); and n is the constant of adsorption intensity.

The changes of various parameters in the adsorption process are clarified by determining the adsorption thermodynamic energy. The entropy change (ΔS^0^), enthalpy change (ΔH^0^) and Gibbs free energy change (ΔG^0^) in thermodynamics can be derived from the Van ‘t Hoff and the Gibbs free energy equations [36]:(5)$$K_{D}=\frac{q_{e}}{C_{e}}\;{\mathrm{Van}}^{‘}t\;\mathrm{Hoff}\;$$
(6)$${\ln\;K}_{D}=\frac{\Delta S^{0}}{R}-\frac{\Delta H^{0}}{\mathrm{RT}}\;{\mathrm{Van}}^{‘}t\;\mathrm{Hoff}\;$$
(7)$$\Delta G^{0}=\Delta H^{0}-T\cdot \Delta S^{0}\;\mathrm{Gibbs}\;\mathrm{free}\;\mathrm{energy}\;$$
where, K_D_ is the distribution constant when the reaction reaches equilibrium, L g^−1^; q_e_ is the equilibrium adsorption capacity of DA/*β*-CD, mg g^−1^; R is the ideal gas constant (R = 8.314 J mol^−1^ K^−1^); T is absolute temperature, K; C_0_ is mg L^−1^; ΔH^0^ is enthalpy change, kJ mol^−1^; ΔS^0^ is entropy change, J mol^−1^; and ΔG^0^ is Gibbs freedom Energy, kJ mol^−1^.

## 3. Results and Discussion

### 3.1. Reaction Mechanism and Discussion

The synthesis process of the DA/*β*-CD composite was shown in Figure 1. Firstly, the acylation happened between *β*-CD and p-toluenesulfonyl chloride [37], in which the hydrogen atom on the hydroxy group of *β*-CD was replaced by p-toluenesulfonyl chloride and produced Ts-*β*-CD. The pyridine was employed as a solvent and acid-binding agent during the reaction process, that not only accelerated the speed of the acylation reaction, but also prevented the formation of chlorinated alkanes. Since the addition of Ts-Cl was exothermic, the addition of it was at a slow speed, and the temperature of the ice water bath was controlled at 2–4 °C.

In the surface amination of DA, the APTMS was chosen as the silane agent. The APTMS has organic and inorganic features. Thus, the organic matrix-APTMS-inorganic matrix bond could be formed. The alcoholysis reaction has happened to APTMS, and the covalent bond formed between the (MTO)_3_Si group with the silanol functions of the DA surface [38]. After that, the amino group was stably “grafted” on the surface of DA. The reaction between amino and Ts-*β*-CD was belonged to nucleophilic substitution, and the nonprotonic environment was needed [39]. DMF is a kind of dipolar aprotic solvents with excellent stability and solubility, that was chosen as solvent for the reaction. The reaction was carried out at 60 °C to avoid possible side reactions.

### 3.2. Characterization

The NMR hydrogen spectrum of Ts-*β*-CD is shown in Appendix A. ^1^H-NMR (D_2_O, 500 Hz): δ = 2.21 (s, 3H), 4.70 (s, 13H), 5.00 (s, 7H), 7.22–7.63 (dd, 4H), 3.78 (m, 7H), 3.55–3.75 (m, 21H). The single heavy peak with a chemical shift at 2.21 was assigned to the H of CH_3_ in p-toluene sulfonyl chloride, and the peak with chemical shift at 4.70 was assigned to the H of OH in *β*-CD.

The FT-IR spectra of DA, *β*-CD, and DA/*β*-CD were used to determine the surface group and shown in Figure 2. The obvious absorption peaks of *β*-CD were observed at 3409 cm^−1^, 2928 cm^−1^, 1163 cm^−1^ and 1036 cm^−1^. The broad peak at 3409 cm^−1^ was assigned to the stretching vibration of the hydroxyl group, the peak at 2928 cm^−1^ was due to the stretching vibration of the -CH_2_ group, and the peaks at 1163 cm^−1^ and 1036 cm^−1^ were attributed to the stretching vibration of C-O-C group [40]. As for DA, the intense peak at 1099 cm^−1^ was assigned to the stretching vibration Si-O group. After modification of *β*-CD, the feature peaks are obtained inherited from DA and *β*-CD with little red-shift, that due to the change of vibration mode of the C-OH group after cross-linking *β*-CD on DA.

The surface morphology and microstructure of DA and DA/*β*-CD were determined by scanning electron microscopy (SEM), and the corresponding images are shows in Figure 3. The DA shown a disc-like shape with a diameter of nearly 40 μm (Figure 3a), in which pores with bigger diameters (400–800 nm) are distributed in the central area, whereas the pores with small size (100–200 nm) were periodically distributed on the framework of DA (Figure 3b). That indicated the DA is a kind of microporous materials [41]. After the modification of *β*-CD, the morphological images of DA/*β*-CD were presented in Figure 3c,d. There was nearly no obvious difference in surface orthography between DA and DA/*β*-CD. Furthermore, the pore structures were kept in good condition, that provide enough bonding sites for DPA adsorption.

The XRD patterns of DA and DA/*β*-CD were presented in Figure 4. The obvious diffraction peaks of the DA and DA/*β*-CD were observed at 21.6°, 26.8°, 29.5° and 35.9° (2θ), due to the reflection of silica. The new peaks were observed at 14.2° and 43.2° from DA/*β*-CD compared with DA, which proved that *β*-CD was grafted to the surface of DA [42]. The XRD patterns indicate weaker crystallization and a larger proportion of amorphous phase for DA/*β*-CD. The amorphous phase could improve the permeability of the target molecule, and bring higher adsorption capacity.

To further determine the surface and porous features of DA and DA/*β*-CD, the nitrogen adsorption–desorption isotherms are shown in Figure 5. The adsorption–desorption curves of DA and DA/*β*-CD presented an H3 hysteresis loop and belonged to a type-IV isotherm. DA is an ideal hierarchical framework material with pores distributed nearly between 1nm to 500 nm [43]. The BET surface area of DA was 20 m^2^ g^−1^, and the pore size was mainly distributed at 1.5 nm and 20–80 nm (Figure 5a). After grafting *β*-CD, the BET surface area of DA/*β*-CD was decreased to 16 m^2^ g^−1^, and the amount of pore at 1.5 nm was obviously decreased. These results indicated the porous structures of DA/*β*-CD and the *β*-CD were successfully modified in the pore of DA.

### 3.3. Adsorption Properties

#### 3.3.1. Standard Curve Plotting

A standard aqueous solution (100 ppm) of DPA was prepared as follows: DPA (10 mg) was dissolved in 100 mL water, and then 20, 40, 60, and 80 ppm aqueous solution of DPA were prepared by diluting the standard solution. The absorbance of DPA solution with different concentrations was measured at λ _max_ = 276 nm to, the concentration of DPA used as abscissa and the absorbency used the y value. The linear relationship was plotted, and the standard curve was presented in Appendix A, in which the regression equation was A = 0.01049C + 0.03732 (R^2^ = 0.98089).

#### 3.3.2. Adsorption Performance of DA/*β*-CD

The UV-vis absorption spectra were used to determine the adsorption performance of DA and DA/*β*-CD composites. The residual concentration of DPA in the wastewater was calculated as the equation provided in Appendix A after measuring the UV-vis spectra. As shown in Figure 6, the intensity of the UV-vis spectra was decreased after the addition of adsorbent. As for the DA, only a weak variety was observed from the UV-vis spectra (Figure 6a), and the adsorption capacity toward DPA was 1.3 mg g^−1^. The surface of DA was dominated by silicone hydroxyl groups, that has weak interaction with DPA molecule, and the adsorption mainly relied on the capillary action of the porous structure. In contrast, obvious adsorption was observed from the DA/*β*-CD composite (Figure 6b), and the adsorption capacity of DA/*β*-CD for DPA was 8.4 mg g^−1^. The adsorption capacity of DA/*β*-CD is comparable with the adsorption capacity of diatomite adsorbent for EDC removing (9.4 mg g^−1^) presented by other researchers [42]. The significant improvement of the adsorption capability of DA/*β*-CD was due to the decoration of *β*-CD. The hierarchically periodic pore structure of DA could provide enough surface area for DPA adsorption. In addition, the *β*-CD used in this study is a cyclic oligosaccharide that has a hydrophilic outer surface and hydrophobic inner cavity. The macrocyclic structure of *β*-CD could embed many molecules through hydrogen bonding, hydrophobic interaction, and electrostatic interaction. After adsorption, the corresponding host–guest inclusion complex formed. The molecular gyration diameter of DPA is 0.67 nm, while the internal size of the wide mouth end of the *β*-cyclodextrin round table structure is 0.78 nm [44], which indicated the DPA could easily enter the interior cavity of *β*-CD. The adsorption process of DPA on DA/*β*-CD can be divided into three steps. Briefly, membrane diffusion step, interlayer diffusion phase [45], and dynamic equilibrium phase. First, the intermolecular hydrogen bond was formed between hydroxyl groups on the outer surface of DA/*β*-CD and the hydroxyl groups of DPA, which led to rapid adhesion of DPA onto the surface of DA/*β*-CD. Then, the DPA diffused from the outer surface to the inner cavity of *β*-CD through hydrophobic action and, forming a host–guest inclusion. Finally, the adsorbent reaches a dynamic equilibrium state of saturated adsorption. The adsorption time between adsorbent and target molecule is critical for removing the harmful ingredients from wastewater in the real cases. An efficient adsorbent should adsorb the target contaminant instantly and reach equilibrium. As exhibited in Figure 6b, the adsorption is nearly completed at 10 min, and there are slightly increase when prolonged the adsorption time. The FT-IR spectra of DA/*β*-CD after adsorbing DPA was shown in Appendix A, and the characteristic peaks were similar with DA/*β*-CD.

Three consecutive DPA (100 ppm) adsorptions were carried out and the results shown excellent repeatability (Figure 6c). In real case, the concentration of EDCs in sediment samples usually less than 20 ppm [46,47]. The adsorption performance of DA/*β*-CD for DPA at lower concentration (10 ppm) was investigated and shown in Appendix A, in which the removal ratio of DA/*β*-CD toward DPA could achieve 59%. These results indicated that the DA/*β*-CD is an efficient adsorbent for removing DPA from water. We compared several representative DPA removing technologies as shown in Appendix A, which shows that adsorption method is the simplest and cost-effective technique for the DPA removing.

#### 3.3.3. Effect of pH

The adsorption efficiency of DA/*β*-CD for DPA was highly related to the pH value of the water system. The pH value of the wastewater was adjusted from 2 to 8 by HCl and NaOH. The interaction between pH value and adsorption efficiency was shown in Figure 7. Since the contaminant DPA is a weak organic acid, the initial aqueous solution presented an acidic feature, and the pH value of the aqueous solution of DPA (100 ppm) was 4.3. As presented in Figure 7, the adsorption capability, and the adsorption ratio toward DPA was firstly increased from 2.0 to 6.0 and then decreased from 6.0 to 9.0. The highest adsorption capability of DA/*β*-CD was 9.6 mg g^−1^ at pH 6.0, and the corresponding adsorption ratio reaches up 38%. The reason was the H^+^ competing with the COOH group (in DPA) at acidic conditions and the active sites of DPA became weaker. As the pH value reaches 6, the DPA exists mainly in molecular form in an aqueous solution [48], that is easily bound to DA/*β*-CD by hydrogen bonding, then the DPA enters the internal cavity of the DA/*β*-CD through hydrophobic interaction. However, the ionization of DPA was happened when pH values higher than 7, in which the DPA exists mainly in the ionic form. The DPA in the ionic state is hard adsorbed by DA/*β*-CD adsorbent, which leads to a decrease in the adsorption performance of DA/*β*-CD.

### 3.4. Isotherm Models

In Langmuir isotherm model, the monolayers of target molecules were arranged on the surface of adsorbent [49], while multilayers formed on non-uniform surfaces in the Freundlich adsorption isotherm [50]. The experimental data for the adsorption isotherm curves obtained by varying the initial concentration of DPA. Different values calculated from the equations are listed in Table 1. In Langmuir isotherm model, R_L_ is the separation constant that can be used to understand the adsorption process, and R_L_ = 1/(1 + bC_e_). The good adsorption effect was obtained as R_L_ value within [0, 1]; the R_L_ = 1 and 0 corresponding to linear and irreversible adsorption, respectively. For the Freundlich isotherm, the good adsorption performance was obtained as the k_F_ ranged from 1–10. As shown in Table 1, the calculated R_L_ values were distributed from 0 to 1, and the k_F_ was 1.5941, which indicated the good adsorption performance of DA/*β*-CD toward DPA.

### 3.5. Thermodynamic Analysis

The thermodynamic parameters of the adsorption process were shown in Table 2. The different values of ΔH^0^ corresponding to different driven forces during the adsorption process [51]. The values of ΔS^0^ in Table 2 are negative, which indicates the adsorption of DPA reduced the degree of freedom of the system. ΔH^0^ < 0 and ΔG^0^ < 0, indicating that the adsorption of DPA by DA/*β*-CD is a spontaneous and exothermic process.

### 3.6. Stability and Reusability of the *DA*/β-*CD*

The stability and regeneration capability of adsorbent materials are important for the application [52]. The adsorption performance of the DA/*β*-CD after 3 cycles was investigated. As shown in Figure 8, the regeneration rate of DA/*β*-CD after 3 cycles could still be higher than 80%. Therefore, the DA/*β*-CD composites used in this study are effective and efficient for DPA adsorption in wastewater.

## 4. Conclusions

In summary, a simple and facile method was proposed to prepare the DA/*β*-CD composite, in which the Ts-*β*-CD was synthesized under a mild condition and used to decorate the surface of DA. The DA/*β*-CD composite shown better adsorption capacity than DA as the host–guest interaction. The application of DA/*β*-CD in adsorption was investigated by efficiently removing DPA from wastewater. The porous structure of the DA and macrocyclic structure of *β*-CD enables the DA/*β*-CD composite to have the capability to rapidly adsorb DPA from wastewater. The adsorption process was almost finished in 10 min with remove ratio of DPA at 38%, in which the adsorption isotherm curves fit the Langmuir (R^2^ = 0.9867) and Freundlich (R^2^ = 0.9748) models. After 3 cycles of application, the regeneration efficiency of DA/*β*-CD was still above 80%. Compared with other technologies for removing DPA from aqueous solution, the DA/*β*-CD composite prepared in this study is a cost-effective, environmentally friendly, and simple adsorbent. The future direction of this research would focus on improving the adsorption capability (uniform DA).

## Data Availability

The data presented in this study are available on request from the corresponding author.

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
