# Peer review of "Facile Synthesis of Diatomite/β-Cyclodextrin Composite and Application for the Adsorption of Diphenolic Acid from Wastewater"

_materials, 2022, doi:10.3390/ma15134588_

Round 1

Reviewer 1 Report

The manuscript "Facile synthesis of diatomite/β-cyclodextrin composite and application for the adsorption of diphenolic acid from wastewater" is well written. This can be considered worth publishing in journal "Materials" after following modifications:

1. The introduction section should be improved with recent related references.

2. Author should provide a separate abbreviation list for better understanding. 

3. Scheme 1. The description of yellow dots, white and other components details should be provided with Figure caption for clearitry.

4. Section 3.2 XRD characterization can be added for crystal structure identification.

5. Reusability test should be performed to check the stability of material. 

6. Comparison of this study with other related literature should be performed in tabular for for performance evaluation. 

Author Response

Dear reviewer

We are greatly appreciated to your insight comments and have considered those comments carefully. We have made revisions to the manuscript. Below is a list of responses to the comments. The modifications are also highlighted in blue in the resubmitted manuscript.

Reviewer #1:

The manuscript "Facile synthesis of diatomite/β-cyclodextrin composite and application for the adsorption of diphenolic acid from wastewater" is well written. This can be considered worth publishing in journal "Materials" after following modifications:

  1. The introduction section should be improved with recent related references.

Answer: Thanks for the suggestion, more references about DPA removing were added in the introduction.

“The oxidation methods include ozone oxidation [16], ferrate oxidation [17], and Fenton and Fenton-like oxidation methods [18, 19], these methods are time consuming and may bring pollution.”

  1. Author should provide a separate abbreviation list for better understanding. 

Answer: Thanks for the suggestion, the abbreviation of chemicals was listed in Table S1.

  1. Scheme 1. The description of yellow dots, white and other components details should be provided with Figure caption for clearitry.

Answer: Thanks for this suggestion, labels were added in Scheme 1.

  1. Section 3.2 XRD characterization can be added for crystal structure identification.

Answer: Thanks for the suggestion, the XRD data was measured and added in the manuscript (Figure 4).

“The XRD patterns of DA and DA/β-CD were presented in Figure 4. The obvious diffraction peaks of the DA and DA/β-CD were observed at 21.6°, 26.8°, 29.5° and 35.9° (2θ), that due to the reflection of silica. The new peaks were observed at 14.2°, 43.2° from DA/β-CD compared with DA, which proved that β-CD was grafted to the surface of DA [42]. The XRD patterns indicates weaker crystallization and larger proportion of amorphous phase for DA/β-CD. The amorphous phase could improve the permeability of the target molecule, and bring higher adsorption capacity.”

  1. Reusability test should be performed to check the stability of material. 

Answer: Thanks for the suggestion, the reusability of the DA/β-CD for adsorbing DPA was added in the manuscript (Figure 8).

The stability and regeneration capability of adsorbent materials are important for the application [52]. the adsorption performance of the DA/β-CD after 3 cycles was investigated. As shown in Figure 8, the regeneration rate of DA/β-CD after 3 cycles could still higher than 80%. Therefore, the DA/β-CD composites used in this study are effective and efficient for DPA adsorption in wastewater.

  1. Comparison of this study with other related literature should be performed in tabular for for performance evaluation. 

Answer: Thanks for the suggestion, we compared several representative DPA removing technologies as shown in Table S2, which shows that adsorption method is the simplest and cost-effective technique for the DPA removing.

Reviewer 2 Report

This is an extensive research, with a lot of experimental work. Thematically, the work is interesting for the researchers and professionals and the proposed manuscript is relevant to the scope of the journal, but only after some modifications and clarification from the Authors.

1.     Abstract must be enriched via valuable results which pave the way for understanding the audiences.

2.     Please improve the scientific rigor of the manuscript. In this regard, the obtained results using all the characterization techniques, for all sample and the corresponding discussion for DA, β-CD and DA/β-CD composite should be presented in the manuscript.

3.     Formatting and quality of the figures should be considerably improved (the plots are not of the same size, some of them are too small (ex. Figure 1)/ large (ex. Figure 5). Please “standardize” the Figures by using the same font, size, color, etc. Figure 2. Please replace “T” by “Transmittance” on y-scale.

4.     Section 3.1. Reaction mechanism and discussion should be supported by results. It is an original, proposed mechanism? Please elaborate.

5.     The equations should be moved in the Experimental part.

6.     There has been no proper discussion on the obtained results or comparing them with the results obtained in other works.

7.     The conclusion section should have the main results in quantitative statements as well. Also, please include the following: new concepts and innovations demonstrated in this study, summary of findings, comparison with findings by other workers, and concluding remark.

Author Response

Dear reviewer

We are greatly appreciated to your insight comments and have considered those comments carefully. We have made revisions to the manuscript. Below is a list of responses to the comments. The modifications are also highlighted in blue in the resubmitted manuscript.

Reviewer #2:

This is extensive research, with a lot of experimental work. Thematically, the work is interesting for the researchers and professionals and the proposed manuscript is relevant to the scope of the journal, but only after some modifications and clarification from the Authors.

  1. Abstract must be enriched via valuable results which pave the way for understanding the audiences.

Answer: Thanks for the suggestion, more information was added in the abstract part.

  1. Please improve the scientific rigor of the manuscript. In this regard, the obtained results using all the characterization techniques, for all sample and the corresponding discussion for DA, β-CD and DA/β-CD composite should be presented in the manuscript.

Answer: Thanks to the suggestion, more discussions of the characterization were added in the manuscript, and the XRD data was measured and added in the manuscript.

  1. Formatting and quality of the figures should be considerably improved (the plots are not of the same size, some of them are too small (ex. Figure 1)/ large (ex. Figure 5). Please “standardize” the Figures by using the same font, size, color, etc. Figure 2. Please replace “T” by “Transmittance” on y-scale.

Answer: Thanks for the suggestion, we have replotted the figures.

  1. Section 3.1. Reaction mechanism and discussion should be supported by results. It is an original, proposed mechanism? Please elaborate.

Answer: Thanks for this comment, the similar reaction was reported in previous research, and the reference was cited in the manuscript. [37]

‘The synthesis process of the DA/β-CD composite was shown in Figure 1. Firstly, the acylation was happened between β-CD and p-toluenesulfonyl chloride [37], in which the hydrogen atom on the hydroxy group of β-CD was replaced by p-toluenesulfonyl chloride and produced Ts-β-CD. The pyridine was employed as a solvent and acid-binding agent during the reaction process, that not only accelerated the speed of the acylation reaction but also prevented the formation of chlorinated alkanes. Since the addition of Ts-Cl was exothermic, the addition of it was at a slow speed, and the temperature of the ice water bath was controlled at 2-4 °C.’

  1. The equations should be moved in the Experimental part. 

Answer:  Thanks for the suggestion, the equations were moved in the Experimental part.

  1. There has been no proper discussion on the obtained results or comparing them with the results obtained in other works.

Answer: Thanks for the suggestion, we compared several representative DPA removing technologies as shown in Table S2, which shows that adsorption method is the simplest and cost-effective technique for the DPA removing.

  1. The conclusion section should have the main results in quantitative statements as well. Also, please include the following: new concepts and innovations demonstrated in this study, summary of findings, comparison with findings by other workers, and concluding remark.

Answer:  We greatly appreciate this helpful suggestion; more quantitative statements were added in the conclusion section.

Reviewer 3 Report

This manuscript matched the scientific scope of Materials journal and can be accepted after major revisions. The observations are listed below:

- I suggest to explain each of your abbreviations the first time it appears in the main text, even it is used in the abstract (it must be explained what "EDC" represents)

-Line 8: Decrease the font size for ‘’cn’’

-Correct the “Citation’’ part. There is a typo at author’s name

-Please check the addresses of the corresponding authors both in the manuscript and in the supplementary file

-Line 24: Add a point after ‘’ public health’’

-L28: Verify the sentence: ‘’4,4-bis (4-hydroxyphenyl) valeric acid, also known as diphenolic acid (DPA), that is an important raw material in organic synthesis and polymer industry’’

-Add a space before the numbering of the references in all manuscript

-Figure 1: Decrease the font size for ‘’C’’ (2-40C; 600C)

-Figure 2: Add the unit of measurement for the Transmittance (y-axis). Also, add a space between ‘’Wavenumber’’ and ‘’cm-1’’ on the x-axis

-Figure 4: Add a space between ‘’Relative pressure’’ and ‘(P/P0)’’; ‘’Quantity adsorbed’’ and (cm3g-1STR)’’. The same observation for ‘’Pore diameter’’ and ‘’nm’’; ‘’Pore volume’’ and ‘’cm3 g-1’’. Also, remove '')'' after ''distribution''

-Please verify the legend of the Figure 5 and Figure S3

-The experimental details should be added in all the adsorption figure captions

-Please add a comparison with the literature

-Thermodynamic and regeneration studies should be performed

-FTIR analysis after adsorption can be included

- Please rewrite the conclusion and report the outcome of the paper more clearly, not a summary thereof. In addition, some quantitative results should be included in the Conclusions section

-Reference 21: Subscript Cu2O

-Reference 29: Superscript Cd2+, Hg2+ and Pb2

Author Response

Dear reviewer

  We are greatly appreciated to your insight comments and have considered those comments carefully. We have made revisions to the manuscript. Below is a list of responses to the comments. The modifications are also highlighted in blue in the resubmitted manuscript.

Reply to the comments of Reviewer:

Reviewer #3:

This manuscript matched the scientific scope of Materials journal and can be accepted after major revisions. The observations are listed below:

  1. I suggest to explain each of your abbreviations the first time it appears in the main text, even it is used in the abstract (it must be explained what "EDC" represents).

Answer: Thanks for the suggestion, all abbreviations and their significances are listed in Table S1.

EDC stands for Endocrine Disrupting Chemical.

  1. Line 8: Decrease the font size for “cn”.

Answer: Thanks for the suggestion, we have corrected this issue.

  1. Correct the “Citation’’ part. There is a typo at author’s name.

Answer: Thanks for the suggestion, we have double checked the references.

  1. Please check the addresses of the corresponding authors both in the manuscript and in the supplementary file.

Answer: Thanks for the suggestion, we have corrected the addresses of corresponding author in the manuscript and supplementary files.

  1. Line 24: Add a point after “public health”. 

Answer: Thanks for the suggestion, we have corrected this issue.

  1. L28: Verify the sentence: “4,4-bis (4-hydroxyphenyl) valeric acid, also known as diphenolic acid (DPA), that is an important raw material in organic synthesis and polymer industry”.

Answer: Thanks for this comment, we have rewritten this sentence.

“Diphenolic acid (DPA)has been used as raw material to produce functional polymers [5].”

  1. Add a space before the numbering of the references in all manuscript.

Answer: Thanks for the suggestion, we have reformatted the references.

  1. Figure 1: Decrease the font size for “℃” (2-40 ℃; 60 ℃).

Answer: Thanks for the suggestion, we have corrected this issue.

  1. Figure 2: Add the unit of measurement for the Transmittance (y-axis). Also, add a space between “Wavenumber” and “cm-1” on the x-axis.

Answer: Thanks for the suggestion, we have corrected this issue.

  1. Figure 4: Add a space between “Relative pressure” and “(P/P0)”; “Quantity adsorbed” and “(cm3g-1STR)”. The same observation for “Pore diameter” and “nm”; “Pore volume” and “cm3g-1”. Also, remove '')'' after ''distribution''.

Answer: Thanks for the suggestion, we have replotted Figure 5 and corrected the issues.

  1. Please verify the legend of the Figure 5 and Figure S3.

Answer: Thanks for the suggestion, we have replotted Figure 6 and Figure S4.

  1. The experimental details should be added in all the adsorption figure captions.

Answer: Thanks for the suggestion, the experimental details were added in the adsorption figure captions.

  1. Please add a comparison with the literature.

Answer: Thanks for the suggestion, we compared several representative DPA removing technologies as shown in Table S2, which shows that adsorption method is the simplest and cost-effective technique for the DPA removing.

  1. Thermodynamic and regeneration studies should be performed.

Answer: Thanks to your suggestion, thermodynamic and regeneration studies were performed. The data were added in the manuscript.

‘The thermodynamic parameters of the adsorption process were shown in Table 2. The different values of ΔH0 corresponding to different driven force during the adsorption process [51]. The values of ΔS0 in Table 2 are negative that indicated the adsorption of DPA reduced the degree of freedom of the system. ΔH0<0 and ΔG0<0, indicating that the adsorption of DPA by DA/β-CD is a spontaneous and exothermic process.’

 ‘The stability and regeneration capability of adsorbent materials are important for the application [52]. the adsorption performance of the DA/β-CD after 3 cycles was investigated. As shown in Figure 8, the regeneration rate of DA/β-CD after 3 cycles could still higher than 80%. Therefore, the DA/β-CD composites used in this study are effective and efficient for DPA adsorption in wastewater.’

  1. FT-IR analysis after adsorption can be included.

Answer: Thanks to your suggestion, we have added the FT-IR spectrum of DA/β-CD after adsorption of DPA was shown in Figure S3.

“The FT-IR spectra of DA/β-CD after adsorbing DPA was shown in Figure S3, and the characteristic peaks were similar with DA/β-CD. “

  1. Please rewrite the conclusion and report the outcome of the paper more clearly, not a summary thereof. In addition, some quantitative results should be included in the Conclusions section.

Answer:  We greatly appreciate this helpful suggestion; more quantitative statements were added in the conclusion section.

  1. Reference 21: Subscript Cu2

Answer: Thanks for the suggestion, we have corrected this issue.

Devaraj, M.; Saravanan, R.; Deivasigamani, R.; Gupta, V. K.; Gracia, F.; Jayadevan, S.,Fabrication of novel shape Cu and Cu/Cu2O nanoparticles modified electrode for the determination of dopamine and paracetamol. J. Mol. Liq. 2016, 221, 930-941, doi:10.1016/j.molliq.2016.06.028.

  1. Reference 29: Superscript Cd2+, Hg2+ and Pb2+.

Answer: Thanks for the suggestion, we have corrected this issue.

Bresson, C.; Menu, M.-J.; Dartiguenave, M.; Dartiguenave, Y., Triethoxysilyl-substituted aminoethanethiol ligands for zinc and cadmium complexes and aminoethanethiol-modified silica gel. Evaluation of the corresponding supported molecular trap for metallic pollutant uptake (Cd2+, Hg2+ and Pb2+). Journal of Environmental Monitoring 2000, 2, (3), 240-247, doi:10.1039/B001408G.

Round 2

Reviewer 1 Report

The authors have revised accordingly 

Author Response

Thanks for your suggestions for the original version. We appreciate your satisfactory comment for this revised version.

Reviewer 2 Report

The authors took into account all my concerns. I think in the present form it can be accepted for publication.

Author Response

(The authors gave the same response as above.)

Reviewer 3 Report

1. Correct the “Citation’’ part. There is a typo at author’s name (Pg. 1, left)

Citation: Houa, M.; Wanga, Z.; Yua, Q.; Konga, X.; Zhangb, M. Facile synthesis of diatomite/β-cyclodextrin composite and application for the adsorption of diphenolic acid from wastewater. Materials 2022, 14, x. https://doi.org/10.3390/xxxxx

Please remove ‘’a’’ and ‘’b’’

2. Please add a comparison between the adsorption capacity of your proposed material with the adsorption capacity of other materials presented in the literature

Author Response

Dear reviewer

We are greatly appreciated to your insight comments and have considered those comments carefully. We have made revisions to the manuscript. Below is a list of responses to the comments. The modifications are also highlighted in blue in the resubmitted manuscript.

1. Correct the “Citation’’ part. There is a typo at author’s name (Pg. 1, left)

Citation: Houa, M.; Wanga, Z.; Yua, Q.; Konga, X.; Zhangb, M. Facile synthesis of diatomite/β-cyclodextrin composite and application for the adsorption of diphenolic acid from wastewater. Materials 2022, 14, x. https://doi.org/10.3390/xxxxx

Please remove ‘’a’’ and ‘’b’’

Answer: Thanks for the suggestion, we have corrected this issue.

2. Please add a comparison between the adsorption capacity of your proposed material with the adsorption capacity of other materials presented in the literature.

Answer: Thanks for the suggestion, more discussion was added and the corresponding reference was cited.

 'The adsorption capacity of DA/β-CD is comparable with the adsorption capacity of diatomite adsorbent for EDC removing (9.4 mg•g-1) presented by other researchers [42]. '